# Agility and Industry 4.0 Implementation Strategy in a Quebec Manufacturing SME

**Stéphanie Bouchard \*, Georges Abdulnour \*** and **Sébastien Gamache**

Department of Industrial Engineering, University of Quebec in Trois-Rivières, Trois-Rivières,
QC G8Z 4M3, Canada; sebastien.gamache@uqtr.ca
\* Correspondence: stephanie.bouchard2@uqtr.ca (S.B.); georges.abdulnour@uqtr.ca (G.A.)

**Abstract:** The emergence of new information technologies, market globalization, climate change, labor shortages, and changing consumer habits have led to dynamic demand and the need for customized mass production systems. This has forced companies, especially small- and medium-sized enterprises (SME), to rethink their product design, production, and marketing models to remain competitive by increasing their agility in the face of growing and changing demand. It becomes relevant to investigate how to move efficiently towards customized mass production in an Industry 4.0 (I4.0) environment. The objective of this research is to develop a strategy for implementing I4.0 in manufacturing SME, based on lean, agility, and intelligent modular product design. A literature review made it possible to target the following performance measures: reducing inventory, minimizing makespan, and reducing time to consumer (reaction time). A case study was conducted in an SME in the agri-food sector to validate the proposed strategy. Inventory levels were reduced by more than 70% and time was cut by almost 65%.

**Keywords:** modular structure; dynamic demand; standardization; dynamic cells; Quebec manufacturing SME; customized mass production; agility; connectivity; industry 4.0

## 1. Introduction

With the arrival of the fourth industrial revolution and new technologies such as the Internet of Things (IoT), artificial intelligence (AI), the cloud, and cyber-physical systems, [1] a change in consumption habits has been observed. This new reality is pushing companies to become more agile and connected to stay competitive and meet consumer demand. To achieve this, product design, manufacturing, and marketing methods must be reviewed. Industry 4.0, primarily based on interconnectivity and implemented through digital transformation, is forcing companies to rethink the way they do business to meet this new dynamic demand associated with customized mass production [2].

Among the many markets affected by these transformations is the agro-food sector, which is grappling with dynamic demand, growing competition, and the emergence of customized mass production. In fact, there has been a noticeable decrease in the number of small farms in Quebec in recent years, and the remaining farms are getting larger [3]. Companies that previously dealt with small farms are being forced to expand their market to seek new customers in European and Asian countries. However, this new market is largely focused on Industry 4.0 [4]. With the current labor shortage, the globalization of markets, increasingly specific needs, and the emergence of a need for mass customization, Quebec manufacturing SME that want to compete in this new era of the fourth industrial revolution must be more agile and connected.

A systematic literature review has made it possible to highlight that no approach has been developed to implement agility in Quebec manufacturing SME. A study was conducted to understand the strengths and weaknesses of Quebec SME in a I4.0 context. This study highlighted lack of resources and technological knowledge of SME. Among

other things, the study suggests that companies that want to make a digital shift should implement lean and continuous improvement strategies, reduce their inventory levels, and move toward standardization and modular product design; agility can then be gained by implementing dynamic production cells [5].

The research aims to demonstrate that the proposed strategy is viable and leads to significant results. This paper proposes an implementation strategy for Industry 4.0 and agility in manufacturing SME to meet customized mass production. To achieve the purpose of this research, a case study was used to validate the implementation strategy. Section 2 of this paper provides a literature review related to the strategies for implementing Agility in SME. Section 3 presents the methodology used for the case study. Section 4 shows the results. Then, Section 5 discusses the benefits and success factors of the proposed strategy. In the last section of this paper, we highlight the work performed.

## 2. Literature Review

A review of the literature shows that little research has been conducted to develop a methodology for implementing I4.0 based on improving agility in manufacturing SME in Quebec. The first part of this section deals with Industry 4.0, agility, and the new requirements generated by this revolution. Since no strategy to implement agility in the context of the I4.0 exists, this research is conducted with the objective of providing such a strategy. The second part deals with standardization and modular product design structure as a mean for achieving mass customization. The last part deals with dynamic cells as a way to implement process agility. A summary of the main elements that emerged from the literature review led to the conceptual framework of the research, focusing on the different performance measures and methodologies used.

### 2.1. Research Methodology

A systematic literature review (SLR) was conducted to analyze the scientific knowledge on agility and I4.0 and what makes it operational. Through the search performed on SCOPUS, it was possible to target several relevant scientific articles, scientific reports, and conference papers related to product modularity, standardization, dynamic cells, and business model, all concepts that enable agility. Figure 1 shows the steps of the methodology and the inclusion and exclusion criteria.

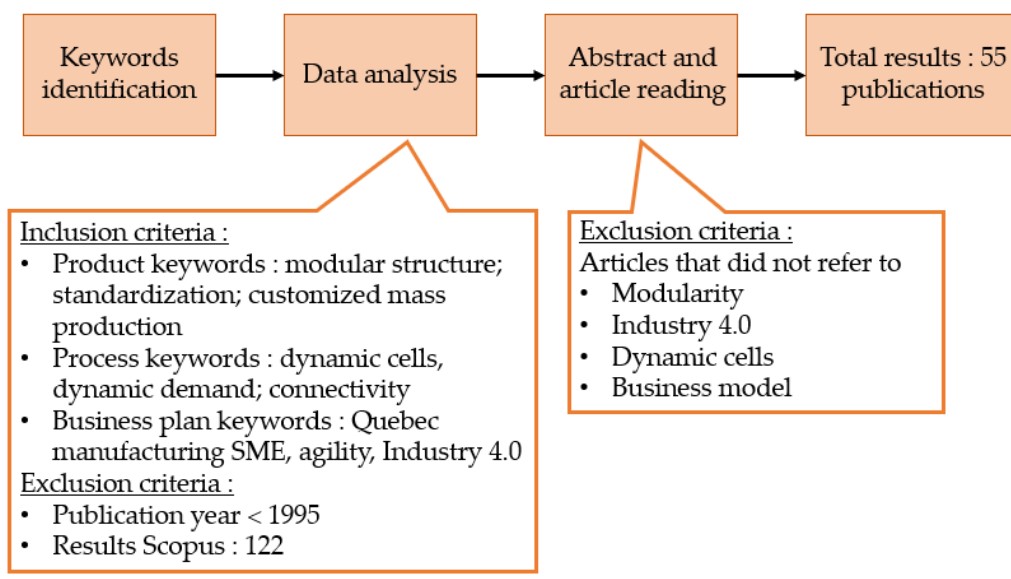

**Figure 1.** Systematic literature review process.

Based on the SLR, Tables 1–6 summarize the authors' main ideas. The tables group, respectively, the dependent and independent variables of the different authors according

to standardization, dynamic cells, and business model in order to better focus the variables discussed. In relation to product standardization, the targeted factors that are studied to determine at what level they have an impact on the inventory level are the standard modules and the similarity index. In relation to the dynamic cells, the targeted factors that are studied in order to know at what level they have an impact on the throughput time are the natural balancing, the assembly sequence, the dynamic cells, the machine flexibility, and the production layout. In relation to the business plan, the targeted factors that will be studied to know at what level they have an impact on the number of catalogue products are predictive maintenance and catalogue sales. Finally, inventory level, throughput time, and number of catalogue products will be the targeted factors that will be studied to determine at what level they have an impact on the company's performance.

**Table 1.** Dependent variables related to standardization according to authors.

| Authors | Reconfiguration Time | Performance | Reaction Time | Time of Passage | Work in Progress | Manufacturing Agility | Inventory Level |
|---|---|---|---|---|---|---|---|
| Gamache, S. [2] | X | ☐ | X | X | X | ☐ | ☐ |
| Egilmez, G. et al. [6] | ☐ | X | ☐ | ☐ | ☐ | ☐ | ☐ |
| Zhang, Y. et al. [7] | X | ☐ | X | ☐ | ☐ | X | X |
| Cohen, Y. et al. [8] | ☐ | X | ☐ | ☐ | ☐ | ☐ | ☐ |
| Frequency | 2 | 2 | 2 | 1 | 1 | 1 | 1 |

**Table 2.** Independent variables related to standardization according to authors.

| Authors | Similarity of Products | Modular Design | Sequence of Operations | Multi-Disciplinarity |
|---|---|---|---|---|
| Gamache, S. [2] | X | X | ☐ | X |
| Egilmez, G. et al. [6] | X | X | X | X |
| Zhang, Y. et al. [7] | X | ☐ | ☐ | ☐ |
| Cohen, Y. et al. [8] | X | X | X | ☐ |
| Frequency | 4 | 3 | 2 | 2 |

**Table 3.** Dependent variables related to the dynamic cells according to the authors.

| Authors | Total Cost | Production Cost | Handling Cost | Reconfiguration Cost | Performance | Cost of Failure | Reconfiguration Time | Lead Time |
|---|---|---|---|---|---|---|---|---|
| Drolet, J. et al. [9] | ☐ | ☐ | ☐ | ☐ | X | ☐ | ☐ | X |
| Gamache, S. [2] | ☐ | ☐ | ☐ | ☐ | ☐ | ☐ | X | X |
| Rheault, M. et al. [10] | X | X | X | X | ☐ | ☐ | ☐ | ☐ |
| Rheault, M. et al. [11] | X | X | X | X | ☐ | X | ☐ | ☐ |
| Safaei, N. et al. [12] | X | X | X | X | ☐ | ☐ | ☐ | ☐ |
| Sakhaii, M. et al. [13] | X | X | ☐ | X | X | X | X | ☐ |
| Niakan, F. et al. [14] | X | X | X | ☐ | X | X | ☐ | ☐ |
| Frequency | 9 | 7 | 5 | 5 | 4 | 4 | 3 | 2 |

**Table 4.** Independent variables related to the dynamic cells according to the authors.

| Authors | Sequence of Operations | Inter/Intra Cellular Arrangement | Machine Capacity | Production System | Turbulent Environment | Production Volume | Production Planning | Multidisciplinarity | Employee Assignments | Machine Reliability |
|---|---|---|---|---|---|---|---|---|---|---|
| Drolet, J. et al. [9] | X | X | X | X | X | ☐ | X | ☐ | ☐ | ☐ |
| Gamache, S. [2] | ☐ | ☐ | ☐ | X | ☐ | X | X | X | X | ☐ |
| Rheault, M. et al. [10] | X | ☐ | ☐ | ☐ | ☐ | ☐ | ☐ | ☐ | ☐ | ☐ |
| Rheault, M. et al. [11] | X | X | ☐ | ☐ | ☐ | ☐ | ☐ | ☐ | ☐ | ☐ |
| Safaei, N. et al. [12] | ☐ | X | ☐ | ☐ | ☐ | ☐ | ☐ | ☐ | ☐ | ☐ |
| Sakhaii, M. et al. [13] | X | X | X | ☐ | ☐ | ☐ | X | X | X | X |
| Niakan, F. et al. [14] | ☐ | ☐ | X | ☐ | X | ☐ | ☐ | X | X | ☐ |
| Frequency | 8 | 6 | 6 | 3 | 3 | 3 | 3 | 3 | 3 | 1 |

**Table 5.** Dependent variables related to the business model according to the authors.

| Authors | Performance | Manufacturing Agility | Efficiency | Flexibility | Ease of Maintenance | Maturity Level |
|---|---|---|---|---|---|---|
| Harris, G. et al. [15] | X | X | X | X | □ | X |
| Moeuf, A. et al. [16] | X | X | X | X | □ | □ |
| Mittal, S. et al. [17] | □ | □ | □ | □ | □ | X |
| Cohen, Y. et al. [8] | X | □ | □ | □ | □ | □ |
| Bourezza, E.M. et al. [18] | □ | □ | □ | □ | X | □ |
| Shashi et al. [19] | X | X | X | X | □ | □ |
| Erez, A.S. et al. [20] | X | □ | □ | □ | X | □ |
| Nishi, T. et al. [21] | X | X | X | X | □ | □ |
| Frequency | 11 | 7 | 6 | 6 | 3 | 3 |

**Table 6.** Independent variables related to the business model according to the authors.

| Authors | 4.0 Technology | Predictive Maintenance | Catalogue Sales |
|---|---|---|---|
| Harris, G. et al. [15] | □ | □ | □ |
| Moeuf, A. et al. [16] | X | □ | □ |
| Mittal, S. et al. [17] | X | □ | □ |
| Cohen, Y. et al. [8] | □ | X | □ |
| Bourezza, E.M. et al. [18] | X | X | X |
| Shashi et al. [19] | X | □ | □ |
| Erez, A.S. et al. [20] | X | X | X |
| Nishi, T. et al. [21] | X | □ | X |
| Rauch, E. et al. [22] | X | □ | □ |
| Frequency | 13 | 4 | 4 |

*2.2. Industry 4.0*

To date, few methods for increasing Quebec manufacturing SME agility appear in the scientific literature. Increased agility makes it possible to meet mass custom demand in the era of the I4.0 revolution. Industry 4.0 was first put forward in 2011 at the Industrial Technology Fair in Hannover, Germany. This fourth industrial revolution is Germany's response to the emergence of technology in the industrial environment and the government's desire to remain the global leader in manufacturing [23,24]. This industrial policy was initially aiming at maintaining a global competitive advantage in the manufacturing business sector [8]. Industry 4.0 relies, among other things, on the connectivity of technological resources for more efficient use of production data [25]. Furthermore, Industry 4.0 generates notable benefits that leads to increased productivity, increased customer satisfaction, and improved ability to innovate [26].

Industry 4.0 generates a globalization of markets and customized mass production that forces companies to review their business model as well as their production methods to remain competitive. The entire process, from sales to after-sales service, must be reviewed and adapted to this new reality and to the growing demands of customers. Therefore, process automation and the implementation and appropriation of digital technologies can increase agility, efficiency, and performance [5,22]. Such an increase in agility, efficiency, and performance allows for greater responsiveness in the context of mass customization demand. In seeking to increase connectivity and automation in manufacturing, the fourth industrial revolution attempts to integrate technological resources to existing production processes. However, it can be observed that small and medium-sized enterprises have greater difficulty harnessing the potential of Industry 4.0 [22]. Rauch et al. [22] focused on the prerequisites for and barriers to smart manufacturing implementation in SME. Among the different prerequisites that emerge, four are considered significant. These are agility and mass customization, real-time data and connectivity, advanced manufacturing and automation, and ease of use of technologies. Agility and mass customization seek to build a manufacturing system that is quickly adaptable and reconfigurable in the face of rapid and constant volume and product changes. Agility makes it possible to meet mass customization demand [22].

The fourth industrial revolution requires greater production flexibility, agility, and efficiency. Due to the growing competitiveness in manufacturing markets, companies are forced to make their production systems more productive and flexible. A production system with a high degree of agility can better respond to changing demand. Similarity

in the production processes of different products also promotes an adapted system [14]. Finally, in their work, Niakan et al. [14] asserted the relevance of making use of dynamic cells as a production system to be able to achieve the requirements of Industry 4.0.

In the context of Industry 4.0, it is important to have a maintenance system given the significant amount of mechanical and electronic equipment. While it is necessary to monitor the production equipment in place, it is also essential to generate information to detect failures so that action can be taken in real time [18]. In their work, Bourezza and Mousrij [18] sought to implement an intelligent industrial maintenance platform that can acquire data in real time to detect breakdowns but also to be able to estimate the time before a device's life span is reached. This information allows for better equipment monitoring and reduces reaction time when a breakdown occurs. In a similar vein, Erez et al. [20] investigated the design methodology of different sensors in the field of nanoscience for industrial machines in the context of predictive maintenance in an Industry 4.0 context [20]. This research highlights the importance of using sensors to enable predictive maintenance of different devices when seeking to reduce reaction times and production process performance.

After-sale service is also crucial in the context of Industry 4.0. Cimini et al. [15] focused on digital servitization and competence development. Servitization is a trend that can be found in industry to offer customers solutions that integrate different goods, services, support, and knowledge by making use of new technologies to facilitate innovation. A product can therefore be used to sell the result of a service. They proposed a model oriented towards selling a solution instead of selling a product. Based on the strategies used and the capabilities of the products, this model makes it possible to situate a company and establish an action plan according to the company's objective [27].

In their work, Nishi et al. [21] discussed the use of electronic catalogues to build a virtual production chain in a I4.0 context. The transition to the virtual world allows companies to study the different opportunities available more easily as well as the possibilities of expansion [21]. The use of online sales catalogues facilitates the implementation of a virtual production chain, which in turn makes it easier to simulate production phase planning. For this purpose, online catalogue models specific to the type of business are implemented. The work provides a method for configuring the production chain based on electronic catalogues [21].

A study by The Boston Consulting Group [28] of more than 800 respondents shows that only 30% of digital transformations are successful. In the remaining 70% of cases, transformations fail to meet targeted goals and expectations in terms of people, processes, infrastructure, sustainability, and ability to innovate within a given time horizon. Such a high failure rate is even more concerning in a context where digital transformation is at the heart of the economic changes taking place. According to The Boston Consulting Group (BCG) [28], six key factors could increase the chances of a successful digital transformation from 30% to 80%: an integrated strategy with clear transformation goals, executive commitment, the use of modular data technology, agile governance to respond quickly and flexibly to demand, talent deployment, and effective progress monitoring.

With their focus on responsiveness and flexibility, SME offer an environment for developing the concept of agility. As a start, Shashi et al. [19] discussed the concept of enterprise supply chain agility as a critical strategy for developing flexible capabilities to respond quickly to changing customer demands. Supply chain agility, defined by Goldman et al. [29], is a strategy for being responsive and ready for change in the face of dynamic and turbulent demand. Harris et al. [15] conducted interviews with various manufacturing SME with the goal of identifying the gaps between companies' digital manufacturing. In their research, the authors discovered that the manufacturing SME interviewed have a low production volume but a wide variety of products, making it important to be agile in the face of changing demand. In addition, Mittal et al. [17] present the characteristics of a manufacturing SME in the context of Industry 4.0. The authors depict SME as one of the strengths of the manufacturing economy and as the guiding light of the manufacturing industry. The fourth industrial revolution has a far-reaching impact

on SME. Furthermore, the article explains that SME often face significant challenges in implementing new technologies. As a result of their research, they conclude that:

-   SME have excellent relationships with their customers;
-   SME are highly dependent on their collaboration networks;
-   SME often have limited knowledge in specific areas;
-   -SME have a greater variety of products that can be adapted to demand;
-   SME have a less complex and more informal organizational structure.

The authors note a major difference between SME and larger companies. This paper defines an SME as an enterprise with less than 250 employees and less than CAD 50 million in annual sales.

For safety concerns, Fujdiak et al. [30] addressed the relationship between safety, cybersecurity, and performance. Therefore, basic security requirements are brought forward in this study with examples of the impact of security on performance. Furthermore, in another study, Fujdiak et al. [30] looked at the security and performance of data distribution service as well as the logic behind it.

I4.0 in SME is increasingly being studied. Studies show that there is a gap in the ways to implement I4.0 at the research level. Da Silva et al. [31] studied the scientific contributions related to the deployment of I4.0 in companies. They found that different issues can hinder the implementation of Industry 4.0. These include lack of financial resources and operational and organizational structure. However, the study allows us to establish requirements that can facilitate the implementation of I4.0 and target the benefits of operationalizing I4.0 [31].

Sony and Naik [32] examined how I4.0 can be successfully implemented in companies. Their systematic literature review identified the following 10 critical factors for successful implementation of I4.0: align Industry 4.0 initiatives with organizational strategy; senior management must fully support the project; employee participation will be important to the success of Industry 4.0; make products and services smart; working to digitize the supply chain; turning the company digital; change management; project management; Industry 4.0 and sustainability; and cybersecurity management.

Although I4.0 has become a concept adopted by a growing number of companies, several studies put forward the difficulty managing the transition to I4.0, while others are even reluctant to advance it [33]. Hoyer et al. [33], therefore, sought to establish a knowledge base to guide future research and to be able to link what has already been studied and highlight perspectives to allow for a better understanding of the complexity of I4.0. In this study, 14 factors for successfully implementing I4.0 are detailed. They are political support, standardization, and IT security; cooperation between companies and institutions; assessment of costs and available funding options; available knowledge and education; pressure to adapt; perceived benefits of implementation; strategic consideration; IT infrastructure maturity; internal knowledge and skills development; lean manufacturing experience; occupational health and safety; industrial sector; and company size. They concluded that the different factors are interrelated and that some are of greater importance and should be prioritized. However, more research is needed to assess the importance of factors in different environments [33].

In their work, Davies et al. [34] reviewed the infrastructure of I4.0 and highlighted that the operationalization of Industry 4.0 is justified beyond cost and efficiency gains. Indeed, mobilizing internal resources makes it possible to create a competitive advantage. The study details the socio-technical requirements to successfully implement I4.0. The article concludes that it is essential to deploy an initiative based on the approval of the socio-technical characteristics put forward [34].

The authors agree on the relevance of increasing the agility, flexibility, and performance of existing production processes to respond to customized mass demand and to be competitive in a context of market globalization. It is important to develop a formal method to increase agility and reduce a company's reaction time in such an environment. While the concept of Industry 4.0 is broad and includes many elements, the current work

defines the fourth industrial revolution as the interconnectivity of emerging technologies to increase agility and interconnectivity of existing industrial systems and processes to meet dynamic demand.

### 2.3. Standardization and Modular Structure

The concept of standardization is particularly important in industrial domains with rather heavy machinery due to the need to transport, assemble, and disassemble the machinery, as explained by Zhang et al. [7]. The authors demonstrated in their research the relevance of using modules that can be configured and reconfigured. This module standardization is achieved by designing modules with independent functions that can be joined using connection interfaces. The authors demonstrated that a final assembly is the result of a combination of different modules that can be used to design a variety of products that can meet customized demand. It is simply a matter of replacing certain functional modules to obtain a different structure. Finally, they concluded by explaining that such a standardized modular design allows for more streamlined structures, easier assembly, longer machine life, and easier maintenance of the equipment.

Egilmez et al. [6] explored the high similarity of manufacturing processes leading to work cell formation. At the end of their research, they concluded that there is a link between good work cell formation and a high similarity index of the cell equipment. Standardization is therefore a key aspect of successfully implementing dynamic work cells.

Gamache's [2] research focused on the key success factors for implementing dynamic cells in network enterprises. The goal of his research was to determine the effects of using dynamic cells and modular design equipment on the performance of network enterprises. Using an experimental design, simulation, and a case study, the author was able to validate the relevance of his model. Gamache [2] concluded that standardization through modular-structured equipment allows for the successful implementation of dynamic cells. The author found that the implementation of dynamic cells optimizes the performance of companies. Thus, the standardization of modules, the implementation of dynamic cells, and the performance of companies are, according to the author, intimately linked. Nevertheless, the author did not address in a concrete way how to implement this standardization and modular design.

The authors agree on the relevance of standardizing in a modular structure to obtain significant benefits for competitiveness in a context of customized mass demand. However, the literature does not address how to do this and the related impacts on mass customization. It is interesting to pursue this line of research with a greater focus on the level of standardization and the impact on agility and response time in a small to medium-sized manufacturing company.

### 2.4. Dynamic Cells

The dynamic cell concept was put forward in 1995 by Rheault et al. [5,11] to fit the reality of subcontracting companies that wanted to produce a wide variety of parts for different customers. To operate in such an environment and maintain their level of competitiveness, companies must rely on flexibility. These authors were the first to use the term dynamic cells, commonly referred to in the scientific literature as dynamic cellular manufacturing systems (DCMS). Dynamic cells represent a production method by physically grouping different workstations or machines, which can be assembled and disassembled several times according to demand, to manufacture a range of products more efficiently with a high index of similarity [2,9]. However, as Rheault et al. [10] explained, moving machines should occur if and only if it is economically justifiable to do so.

According to Rheault et al. [11], the concept of a dynamic cell exists because of the turbulent environment in which the manufacturing system finds itself. The turbulent environment is described by the authors as an environment in which there is a large variation in production batches, quantity of demands, processing times, and setups. According to the authors, an environment is considered turbulent due to the constant changes in products [10].

The authors referred to the work of Ramudhin and Rochette [35], Montreuil et al. [36], Irani [37], Rajamani et al. [38], Hayes and Pisano [39], Greene and Cleary [40], and finally, Kusiak and Heragu [41] to characterize this environment. Furthermore, an environment is characterized as turbulent when there are frequent product changes at the production level but also when the competition on the market is strong. Finally, the authors explained that due to the turbulence of the environment, production lines and other fixed installations quickly become obsolete. Dynamic cells have also been shown to be effective in dynamic environments on several occasions [10].

Drolet et al. [9], for their part, examined the comparison between different production systems in the context of a case study. The following production systems were studied: job shop, classical cells, virtual cells, and dynamic cells. Using an experimental design, the authors were able to draw conclusions about the relevance and performance of a dynamic cell production system. Dynamic cells were found to be superior to other types of production systems for each of the 12 performance factors targeted in this study. These 12 performance factors include total marginal cost, average level of work in progress, maximum level of work in progress, average level of delay, and maximum level of delay, among others [9].

To quantify the gains achieved through dynamic cells, Gamache [2] focused on the effects of dynamic cells on the performance of network firms. By focusing on supplier reliability, the production system, interdisciplinarity, and modular design, among other things, and through an experimental design, the author was able to reach a conclusion on the effect of dynamic cells. He concluded that such a production system is relevant in the context of network enterprises [2]. Furthermore, the author confirmed that the variables that were studied make it possible to optimize the reaction time, the number of WIP, the throughput time, and the total sales. However, the author concluded by offering an opening for future research with a greater focus on factors that were overlooked in his research. These factors include the planning horizon, production scheduling, the study of a more elaborate network, and outsourcing, among others.

The authors agree that to operationalize and implement a standardized design, a suitable manufacturing system is necessary. Due to the variable structure created by modular products, a fixed layout often causes significant inconvenience. As a result, several authors have addressed the notion of dynamic cells in combination with reconfigurable modules [2].

### 3. Methodology

As shown in the literature review, agility and I4.0 are interrelated. Agility makes it easier to implement I4.0, whereas Industry 4.0 makes it possible to increase a company's agility. This implies that the more agile a company is, the more it is able to draw notable benefits. Finally, it can be understood that agility and Industry 4.0 are crucial concepts, allowing both to respond to the needs generated by the context of personalized mass demand. Figure 2 shows the concepts studied and how they relate to each other.

The literature review shows that there is no method to determine how to evolve and compete in a I4.0 environment. As such, the methods depicted in Figure 2 are inspired by Beaudoin et al. [42], who suggested that there are three strategies for implementing Industry 4.0 in a manufacturing company. They show that companies could either work on their product, their process, or their services. We believe that a company aiming to respond to customized mass production need to first address the product. Then, they must adapt their processes to the newly developed product. Finally, services should be improved in line with the new product. This therein suggests an implementation strategy based on adapting the product, the production process, and the business model.

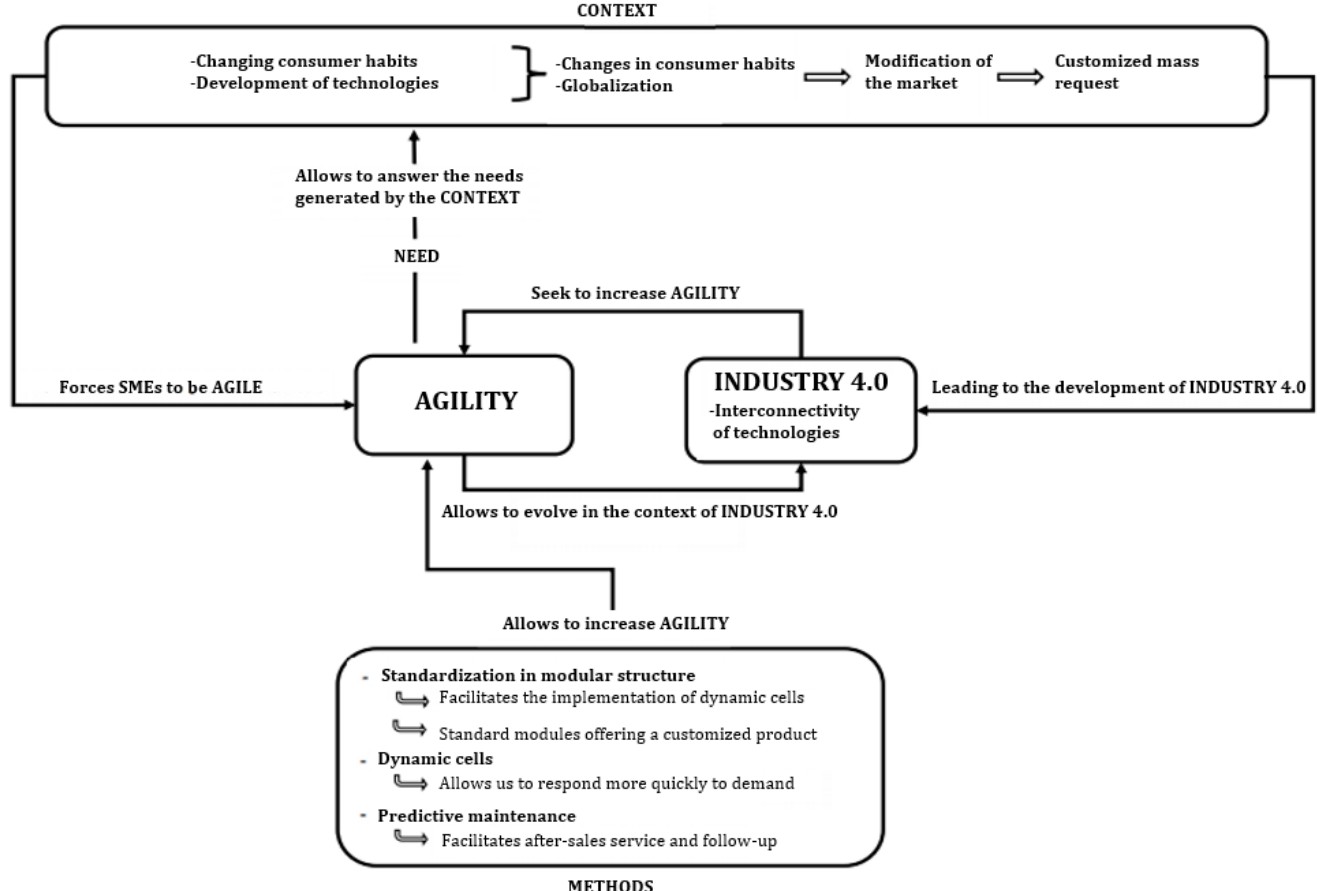

**Figure 2.** Conceptual framework.

In this sense, and based on the literature, standardization and modular design as well as the similarity index are the targeted factors investigated for the product. Second, process agility is reached by dynamic cells, machine flexibility, mixed line assembly, and natural assembly line balancing. Thirdly, services such as predictive maintenance and online catalogue sales will be the other targeted group of factors that will be studied.

A case study is the methodology used in this research to develop and validate the implementation strategy. Although the effectiveness of dynamic cells and the relevance of reconfigurable modules have been demonstrated in the literature, it is interesting, within the framework of this research, to further investigate the method and standardization tools to ensure the successful implementation of dynamic cells to increase the agility of manufacturing SME. The global implementation strategy proposed in Table 7 goes through three consecutive and interrelated steps. The first step addresses standardizing the product to transform it into a modular form. The second step addresses the production line where the linear physical flow and the information flow are promoted to adapt the production process to a I4.0 environment. This is possible by implementing, among other things, a management system and an adapted codification and by reviewing the production system in place. Finally, the third step consists of adapting the business model to make the sales process capable of after-sales service I4.0 by implementing an adapted marketing model and prioritizing predictive maintenance.

**Table 7.** Global implementation strategy.

| Global Implementation Strategy | | |
|---|---|---|
| Objective | To offer a method to increase the agility of a Quebec manufacturing SME and to be able to respond to customized mass demand using Industry 4.0 tools. | |
| How to Proceed | | |
| Step 1 | Make the product 4.0 | Standardization and modular structure<br>1. Revise the design to obtain a high similarity index between the products.<br>2. Standardize and modular structure by platform.<br>3. Develop easily reconfigurable modules. |
| Step 2 | Make the production process 4.0 | Promote linear physical flow and information flow<br>1. Implement an ERP system.<br>2. Implement a coding system adapted to modular structures.<br>3. Implement a Kanban inventory management system.<br>4. Set up dynamic work cells. |
| Step 3 | Adapt the business model to I4.0 | Establish online sales and after-sales service<br>1. Implement marketing focused on online catalogue sales.<br>2. Use connected sensors to facilitate predictive maintenance. |

This study is part of a larger research program having as objective the development and adaptation of strategies of Lean 4.0's successful implementation in SME. Manufacturing enterprises subjects for the case studies were chosen based on their innovation capability, financial stability, and sustainable growth. Four analyses steps were used for selecting an enterprise. As part of a second research program on lean implementation in SME, the first step consisted of a visit by an Innovation agent of NRC (National Research Council of Canada) to the factory. An assessment was made by the agent based on the criteria of innovation, financial stability, and growth possibility. Once an enterprise was selected, the second step consisted of accomplishing a VSM by the researcher to develop a strategy for implementing lean manufacturing. Based on the results of the VSM and the implication and the will of the management staff and the executive direction to participate, in the third step, we assessed the capability and ability of the enterprise to be part of the study as a case study. At the fourth step, we developed a research demand for the Mitacs grant for a master's or Ph.D. student to start the project. The case study became a two-years applied research program. The case study and the research program, at this step, were evaluated by at least two academics researchers. If accepted, the case was realized. Three cases were already completed with great success, and we compare the results in the discussion section. This case study was chosen using the same methodology [43–45].

Each of these steps was implemented in a Quebec manufacturing SME specializing in the design and manufacture of robotic machines for the agri-food sector. Among the products most often sold by the company are feeding robots, mixers, and conveyors. Different types of conveyors are offered, but belt conveyors and feeder conveyors represent the largest percentage of sales. Implementing the strategy in the manufacturing SME made it possible to validate the relevance of this strategy, its viability, as well as its impacts on the system. Furthermore, implementation in the field allowed us to highlight the success factors and obstacles related to this global implementation strategy.

## 4. Step by Step Implementation

### 4.1. Make the Product 4.0

The following analysis, detailed in Tables 8 and 9, identifies in detail each of the options that account for more than 80% of sales related to belt conveyors (CC) and feeder conveyors (CN). This analysis allows us to target the most frequently sold options for both CC and CN. It is easier to eliminate rarely sold options to facilitate the standardization stage.

**Table 8.** Options present more than 80% of the time in CC sales.

| Options | Details | More Than 80% |
|---|---|:---:|
| Drive mechanism | Single 70 series<br>Single 80 series<br>Single 80 series large roll<br>Double series 80 large roll | X<br>X<br><br> |
| Galvanized steel conveyor section | 2.5′ section<br>4′ section<br>5′ section<br>8′ section<br>10′ section | <br><br><br><br>X |
| Angle | Concave<br>Convex | X<br> |
| Mechanical diverter | Lever with cylinder<br>Manuel lever | X<br> |
| Mechanical fastener | Steel<br>Stainless steel | X<br> |
| Edge for open section | 5′ edge<br>10′ edge | <br>X |
| Edge and cover for closed section | 5′ edge<br>10′ edge<br>5′ cover<br>10′ cover | <br>X<br><br>X |
| Hopper | 27″ (standard belt)<br>27″ (paddle belt)<br>48″ extended (standard belt)<br>48″ extended (paddle belt)<br>36″ for double head<br>36″ for double head (standard belt)<br>36″ for double head (paddle belt)<br>Central 41″ with attachment<br>Cover transition<br>Deflector cover | X<br><br><br><br><br><br><br><br><br> |
| Silo bag adapter | 12″ silo bag adapter<br>10″ silo bag cover<br>12″ silo bag cover<br>41″ central hopper 10″ bag<br>41″ central hopper 12″ bag<br>12″ bag outlet adapter<br>12″ silo bag outlet adapter ext. brush | X<br><br>X<br><br><br><br> |
| Options to be installed on the drive head | Spill deflector<br>Scraper for smooth belt<br>Rotary belt cleaning brush (70 series)<br>Rotary belt cleaning brush (80 series)<br>Rotating belt cleaning brush (80 series)<br>Spill plate with magnet | X<br>X<br>X<br><br><br> |

**Table 9.** Options present more than 80% of the time in CN sales.

| Options | Details | More Than 80% |
|---|---|:---:|
| Drive mechanism | Movable plow with left and right deviator (belt 1 or 2 direct)<br>Fixed plow left or right (belt 1 direct)<br>Movable plow with left and right deviator (belt 1 direct)<br>Movable plow with left or right deviator (belt 2 direct)<br>Left/right mechanical dumping selector<br>Dumping selector electric cylinder left/right<br>Small roller<br>Big roller | X<br><br><br><br><br>X<br><br>X |
| Cable | Plastic coated steel cable (sold by the foot) | |
| Mechanical deviator and unloading slab | Fixed with electric cylinder (lateral)<br>Fixed with manual lever (lateral)<br>Fixed with lever and electric cylinder (vertical)<br>Fixed with manual lever (vertical)<br>Unloading slab left side<br>Dumping slab right side | <br>X<br><br><br>X<br>X |

In order to shift to easily reconfigurable and connectable modular structures, it is important to understand the nature of the product and of what it is made. We performed a design study of the two different types of conveyors to decide on the nature of the different modules to be standardized. The different modules for each conveyor made it easier to determine the different platforms to be set up in connection with the dynamic cells to be implemented following the standardization. Figure 3 shows the curved conveyor and the nine modules that can compose it.

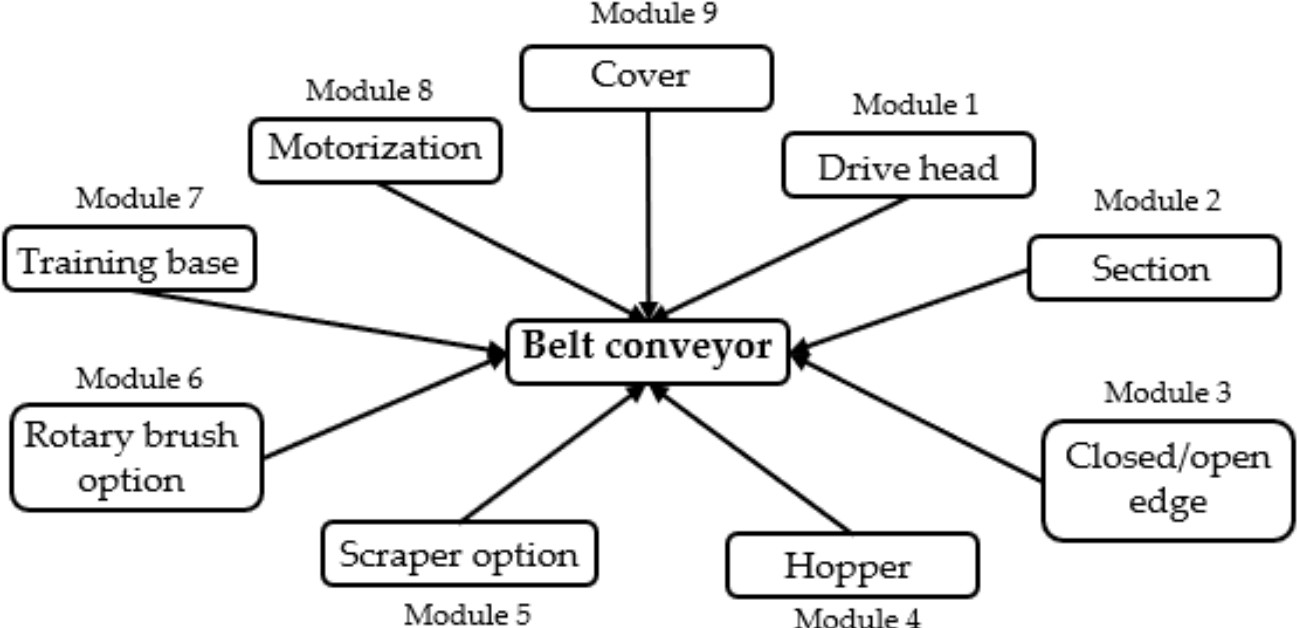

**Figure 3.** Platform 1: CC.

Figure 4 shows the feeder conveyor and the four modules that compose it.

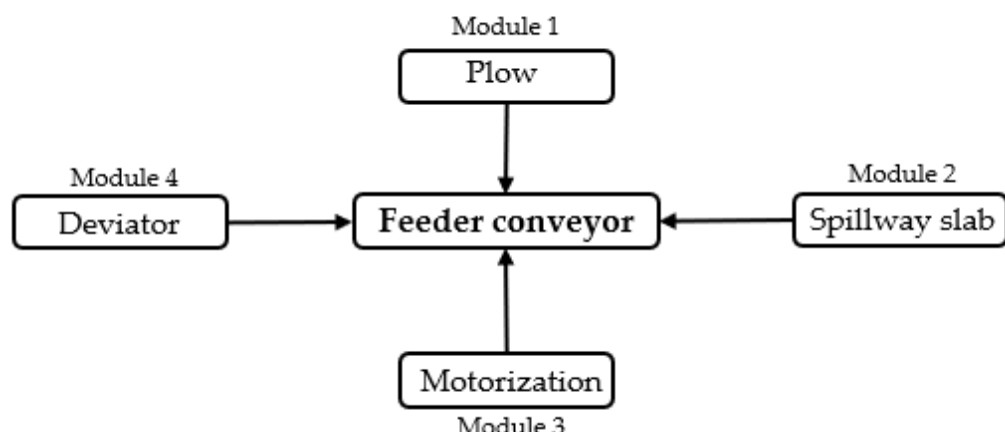

**Figure 4.** Platform 2: CN.

After studying all the conveyors, their options, and the way to subdivide the modules, the module designs were concretized. This step was carried out meticulously in order to take into account as many constraints as possible. Table 10 shows the details of the standardization results for the belt conveyors.

**Table 10.** Standardization of belt conveyors.

| Assembly | Before | After | Percentage Reduction | Notes |
|---|---|---|---|---|
| Drive head | 6 | 2 | 67% | One head model but two tube lengths (18″–24″) |
| Tensioner | 4 | 2 | 50% | One head model but two leg lengths (18″–24″) |
| Section | 20 | 4 | 80% | 10′, 10′ adjustable, 10′ LP, 10′ LP adjustable |
| Edge | 4 | 2 | 50% | 10′ adjustable |
| Cover | 4 | 2 | 50% | 10′ adjustable |
| Hopper | 21 | 2 | 90% | Adjustable bracket (18″–24″), length 48″ and 36″ adjustable to 27″ |
| Brush | 4 | 2 | 50% | Brush length at 18″–24″ |
| Scraper | 2 | 2 | 0% | Rubber length at 18″ and 24″ |
| Total | 60 | 18 | 70% | |

Table 11 shows the details of the standardization results for the feeder conveyors.

**Table 11.** Standardization of feeder conveyors.

| Assembly | Before | After | Percentage Reduction | Notes |
|---|---|---|---|---|
| Plow | 7 | 2 | 71% | Model to try at customers' premises to confirm. |
| Deviator | 6 | 3 | 50% | Removed from inventory. |
| Unloading slab | 6 | 2 | 67% | One making left and right (18″–24″). |
| Total | 19 | 7 | 63% | |

Following the product standardization stage, the modular structures resulted in a 70% reduction in belt conveyor components and a 63% reduction in feeder conveyor components.

*4.2. Make the Production Process 4.0*

In order to improve the information and physical flow of the manufacturing SME under study, it was important to implement the following:

1. Implement an ERP software;
2. Use a codification adapted to the new modular structures;
3. Use an inventory management method such as the Kanban method;
4. Implement dynamic cells to make the production process 4.0. As demonstrated by the literature review. A production process containing these elements will have superior agility and will therefore be better able to respond to dynamic demand.

In order to promote a linear physical flow and to optimize the flow of information, it was necessary to set up means of application in line with the needs and context of the partner company. The production process must be adapted to the needs and reality of the company. The application set up method makes it possible to obtain an agile process to handle the dynamic demand.

4.2.1. Implement an ERP System

Gamache [10] proposed a strategy for developing digital plans to allow manufacturing SME to increase their digital performance. One of the priority projects in this strategy is BOM redesign and system connectivity. Implementing an ERP system therefore enables increased system connectivity. Table 12 was produced by the industrial engineer responsible for implementation at the partner company. It describes the steps involved in implementing the ERP software.

4.2.2. Implement a Coding System Adapted to Modular Structures

Coding for fabricated components and assemblies that is appropriate for modular structures makes it easier to understand and track the structures in place. It is also important to ensure component traceability with the coding initially in place. To do this, the method explained below was used. Based on the parts and assemblies nomenclature, it was possible to set up a codification adapted to the modular structures. Tables 13 and 14 show how each assembly belongs to the different modules.

**Table 12.** ERP implementation.

| | Steps | Details |
|---|---|---|
| Step 1 | Data cleaning | • Inactive items<br>• Duplicates<br>• Not used<br>• Corrections<br>• Standardization of descriptions<br>• Clients-suppliers-employees |
| Step 2 | Filling templates | • Done in parallel with step 1<br>• Cleaning carried out according to the templates<br>• Data transfer |
| Step 3 | Complete detailed project planning | • Complete planning of the next steps<br>• Planning performed with the integrator |
| Step 4 | Process review | • Minimization of customization<br>• Process improvement<br>• Process standardization |
| Step 5 | Training | • Overall theory<br>• Specific theory per person<br>• Practical |
| Step 6 | Partial use of the software | • Validation that everything works at the accounting level<br>• Review of the functioning of the processes |
| Step 7 | Complete migration | • Complete transfer |
| Step 8 | Follow-up | • Support<br>• Corrective action<br>• Adjustment |

**Table 13.** Coding of belt conveyors.

| Parts | Modules | | | | | | | | |
|---|---|---|---|---|---|---|---|---|---|
| | 1 | 2 | 3 | 4 | 5 | 6 | 7 | 8 | 9 |
| CC41-0602_00 | 1 | 0 | 0 | 0 | 0 | 0 | 0 | 0 | 0 |
| CC51-0602_00 | 1 | 0 | 0 | 0 | 0 | 0 | 0 | 0 | 0 |
| CC01-1013_00 | 0 | 1 | 0 | 0 | 0 | 0 | 0 | 0 | 0 |
| CC01-1014_00 | 0 | 1 | 0 | 0 | 0 | 0 | 0 | 0 | 0 |
| CVLP009 | 0 | 1 | 0 | 0 | 0 | 0 | 0 | 0 | 0 |
| CVLP-1000_00 | 0 | 1 | 0 | 0 | 0 | 0 | 0 | 0 | 0 |
| CC01_0380_00 | 0 | 0 | 1 | 0 | 0 | 0 | 0 | 0 | 0 |
| CC41-0333_00 | 0 | 0 | 0 | 1 | 0 | 0 | 0 | 0 | 0 |
| CC51-0333_00 | 0 | 0 | 0 | 1 | 0 | 0 | 0 | 0 | 0 |
| CC41-0343_00 | 0 | 0 | 0 | 1 | 0 | 0 | 0 | 0 | 0 |
| CC51-0343_00 | 0 | 0 | 0 | 1 | 0 | 0 | 0 | 0 | 0 |
| CC41-0301_00 | 0 | 0 | 0 | 0 | 1 | 0 | 0 | 0 | 0 |
| CC51-0301_00 | 0 | 0 | 0 | 0 | 1 | 0 | 0 | 0 | 0 |
| CC41G410 | 0 | 0 | 0 | 0 | 0 | 1 | 0 | 0 | 0 |
| CC51G410 | 0 | 0 | 0 | 0 | 0 | 1 | 0 | 0 | 0 |
| CC41-0311_00 | 0 | 0 | 0 | 0 | 0 | 0 | 1 | 0 | 0 |
| CC51-0311_00 | 0 | 0 | 0 | 0 | 0 | 0 | 1 | 0 | 0 |
| CC41-0704_00 | 0 | 0 | 0 | 0 | 0 | 0 | 0 | 1 | 0 |
| CC51-0704_00 | 0 | 0 | 0 | 0 | 0 | 0 | 0 | 1 | 0 |
| CC41-0705_00 | 0 | 0 | 0 | 0 | 0 | 0 | 0 | 1 | 0 |
| CC51-0705_00 | 0 | 0 | 0 | 0 | 0 | 0 | 0 | 1 | 0 |
| CC41-0413_00 | 0 | 0 | 0 | 0 | 0 | 0 | 0 | 0 | 1 |
| CC51-0413_00 | 0 | 0 | 0 | 0 | 0 | 0 | 0 | 0 | 1 |
| CVLP-0413_00 | 0 | 0 | 0 | 0 | 0 | 0 | 0 | 0 | 1 |

**Table 14.** Coding of feeder conveyors.

| Parts | Modules | | | |
|---|---|---|---|---|
| | **1** | **2** | **3** | **4** |
| CS41-0179_00 | 1 | 0 | 0 | 0 |
| CS51-0179_00 | 1 | 0 | 0 | 0 |
| CS41-0180_00 | 1 | 0 | 0 | 0 |
| CS51-0180_00 | 1 | 0 | 0 | 0 |
| CS41-0181_00 | 1 | 0 | 0 | 0 |
| CS51-0181_00 | 1 | 0 | 0 | 0 |
| CS41-0182_00 | 1 | 0 | 0 | 0 |
| CS51-0182_00 | 1 | 0 | 0 | 0 |
| CVLP-0400_00 | 1 | 0 | 0 | 0 |
| CVLP-0401_00 | 1 | 0 | 0 | 0 |
| CVLP-0402_00 | 1 | 0 | 0 | 0 |
| CVLP-0403_00 | 1 | 0 | 0 | 0 |
| CS41-0359_00 | 0 | 1 | 0 | 0 |
| CS51-0359_00 | 0 | 1 | 0 | 0 |
| CN4G430 | 0 | 0 | 1 | 0 |
| CN4LPG430 | 0 | 0 | 1 | 0 |
| CN5G530 | 0 | 0 | 1 | 0 |
| CN4G435 | 0 | 0 | 1 | 0 |
| CN4LPG435 | 0 | 0 | 1 | 0 |
| CN5G535 | 0 | 0 | 1 | 0 |
| CS40-0185_00 | 0 | 0 | 0 | 1 |
| CS50-0185_00 | 0 | 0 | 0 | 1 |
| CS40-0187_00 | 0 | 0 | 0 | 1 |
| CS50-0187_00 | 0 | 0 | 0 | 1 |
| CVLP-0348_00 | 0 | 0 | 0 | 1 |
| CVLP-0351_00 | 0 | 0 | 0 | 1 |

This codification made it possible to identify where each module belongs and to maintain traceability with the nomenclature initially in place. In this configuration, no assembly and/or part belongs to multiple modules. Each module has its own modular structure based on its functions. It can also be observed that there are very few assemblies for each module in order to reduce part variability as much as possible and to make them standard.

4.2.3. Kanban Method

In this study, different Kanban levels were identified. Table 15 details the Kanban levels.

**Table 15.** Kanban level.

| Level | Type of Kanban |
|---|---|
| 1 | Raw material |
| 2 | Sub-assembly |
| 3 | Module |
| 4 | Finished product |

Levels 1 and 2 relate to manufacturing, while levels 3 and 4 relate to assembly. By increasing the level of the type of Kanban used, reaction time following a customer order can be reduced. Reducing reaction time increases the agility of a manufacturing SME. However, mass customization results in a finished product customized to the customers' needs. With a wide variety of finished product layouts possible due to customization, it is not possible to go to level 4 and have finished product Kanbans in this context. Figure 5 schematizes the level of Kanban types.

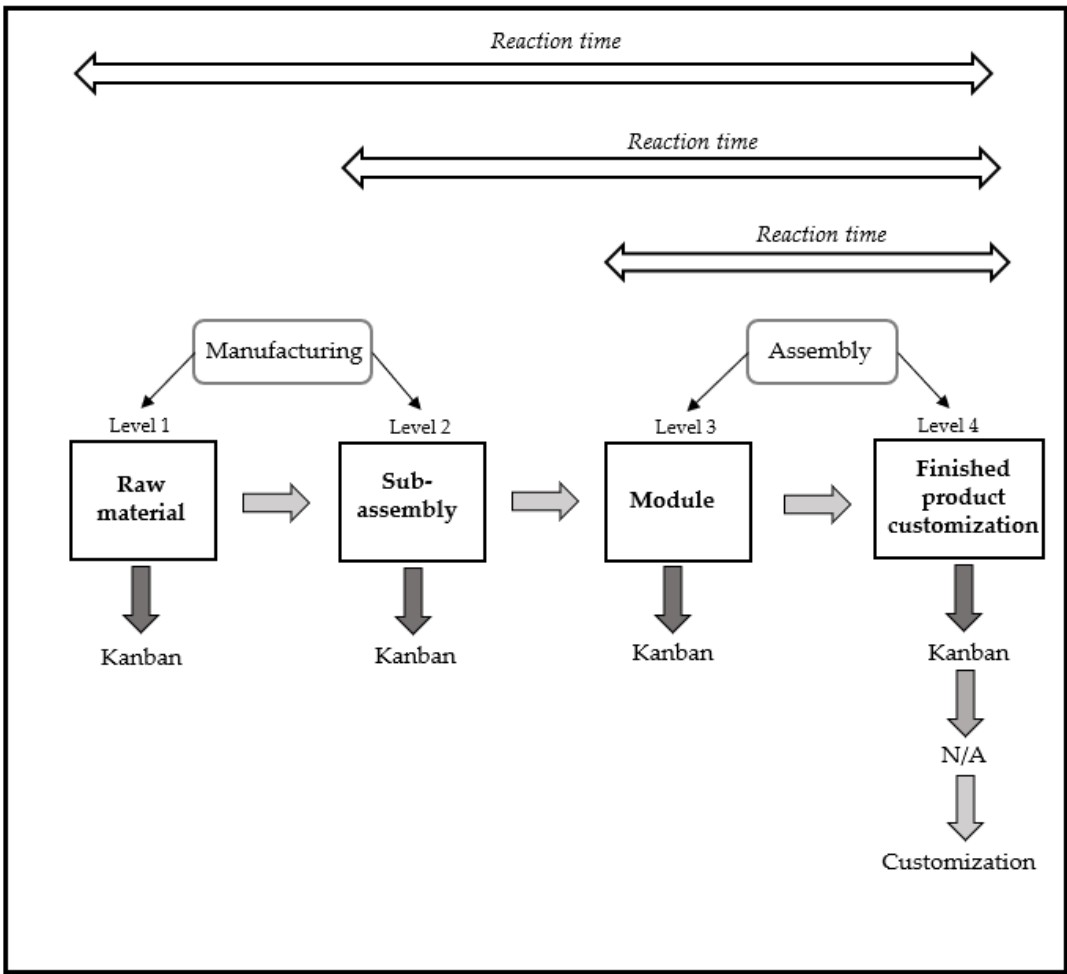

**Figure 5.** Level of Kanban types.

Implementing a level 3 Kanban type is favored in the company studied. Since the production goal is to be able to have two weeks of stock modules in advance production, it is important to achieve a steady state of production to be able to work on orders that are not yet sold. Level 3 Kanbans allow for a significant reduction in throughput time, as shown in Table 16.

**Table 16.** Reduced time to level 3.

| Type of Conveyor | Passage Time (h): Kanban Level 1 | Passage Time (h): Kanban Level 3 | Reduction Percentage |
| --- | --- | --- | --- |
| Belt conveyor | 8.1 | 3.83 | 53% |
| Feeder conveyor | 5.5 | 1.75 | 68% |

By implementing level 3 Kanban types, it is possible to reduce belt conveyor throughput time by 53% and feeder conveyor throughput time by 68%. By reducing the throughput time and the number of parts to be handled due to the reduction of the parts inventory, SME agility increases significantly.

4.2.4. Set up Dynamic Cells

When analyzing the operation times per module for each conveyor, it can be seen that the workstation that requires the most time is assembly (4.27 h for belt conveyors and 3.75 h for feeder conveyors). Furthermore, as mentioned above, this is also the work area that performs tasks in parallel as opposed to the rest of the tasks that are performed in series. Tables 17 and 18 show the details of the times by sector according to the modules.

**Table 17.** Operating time per module of belt conveyors.

| Areas of Work | Total Time (h): Module 1 | Total Time (h): Module 2 | Total Time (h): Module 3 | Total Time (h): Module 4 | Total Time (h): Module 5 | Total Time (h): Module 6 | Total Time (h): Module 7 | Total Time (h): Module 8 | Total Time (h): Module 9 | Total Time (h): |
|---|---|---|---|---|---|---|---|---|---|---|
| Order processing | | | | | 0.2 | | | | | 0.28 |
| Batch programming | | | | | 0.08 | | | | | |
| Cutting | 0.56 | 0.08 | 0.05 | 0.24 | 0.08 | 0.13 | 0.09 | – | 0.07 | |
| Folding | 0.27 | 0.07 | 0.03 | 0.37 | 0.17 | 0.22 | 0.20 | – | 0.12 | |
| Machining | 0.00 | 0.00 | 0.00 | 0.00 | 0.00 | 0.05 | 0.15 | – | 0.00 | 6.01 |
| Welding | 1.00 | 0.00 | 0.00 | 0.00 | 0.00 | 0.00 | 1.08 | – | 0.00 | |
| Painting | 0.53 | 0.00 | 0.00 | 0.00 | 0.00 | 0.08 | 0.37 | – | 0.00 | |
| Assembly | 1.83 | 0.00 | 1.33 | 0.17 | 0.33 | 0.11 | 0.50 | – | 0.00 | 4.27 |
| Installing the transport pallet | | | | | 1.33 | | | | | |
| Inspection | | | | | 1.25 | | | | | 3.83 |
| Pallet loading | | | | | 1.25 | | | | | |
| Passage time (h) | 8.30 | 4.26 | 5.52 | 4.89 | 4.69 | 4.70 | 6.50 | 4.10 | 4.30 | – |

**Table 18.** Operating time per feeder-conveyor module.

| Areas of Work | Total Time (h): Module 1 | Total Time (h): Module 2 | Total Time (h): Module 3 | Total Time (h): Module 4 | Total Time (h) |
|---|---|---|---|---|---|
| Order processing | | 0.2 | | | 0.28 |
| Batch programming | | 0.08 | | | |
| Cutting | 0.75 | 0.10 | – | 0.31 | |
| Folding | 0.79 | 0.05 | – | 0.20 | |
| Machining | 0.30 | 0.47 | – | 0.40 | 4.99 |
| Welding | 0.85 | 0.07 | – | 0.15 | |
| Painting | 0.18 | 0.17 | – | 0.20 | |
| Assembly | 2.50 | 0.25 | – | 1.00 | 3.75 |
| Installing the transport pallet | | 0.67 | | | |
| Inspection | | 0.33 | | | 1.75 |
| Pallet loading | | 0.75 | | | |
| Passage time (h) | 7.40 | 3.14 | 2.03 | 4.29 | – |

The sum of the operation time for the belt-conveyor assembly area is 4.27 h. On the other hand, the sum of the raw material preparation time is 6.01 h and 3.83 h for shipping preparation.

The sum of the operating time for the feeder-conveyor assembly area is 3.75 h. On the other hand, the sum of the preparation of the raw material time is 4.99 h and 1.75 h for preparation for shipping. With the objective of making 250 conveyors per year and considering that we want to spend half the time per week assembling them, i.e., 1000 h per year, it is possible to determine the cycle time resulting from the takt time. The cycle time is 14,400 s per conveyor, i.e., 4 h.

$$\text{Time spent} = \frac{40 \text{ hours}}{1 \text{ week}} \times \frac{50 \text{ weeks worked}}{1 \text{ year}} \times 0.5$$

$$\text{Time spent} = 1000 \text{ hours/year}$$

$$CT = \frac{1000 \text{ hours}}{1 \text{ year}} \times \frac{1 \text{ year}}{250 \text{ conveyors}} \times \frac{60 \text{ minutes}}{1 \text{ hour}} \times \frac{60 \text{ seconds}}{1 \text{ minute}}$$

$$CT = 14400 \sec \text{onds/conveyor}$$

Since the assembly steps can be performed in parallel, there is no precedence for module assembly. Simulating the production start of each module at the same time and taking into account that two stations are available depending on the percentage of module sales, the results in Tables 19 and 20 are obtained for the belt conveyors. A "1" means that the task is performed at the station in question.

**Table 19.** Simulation for belt conveyor station #1 with a 100% sales percentage.

|  | Assembly Station 1 | Assembly Station 2 |
|---|---|---|
| Cost | 1 | 10 |
| Module 1 | 1 | 0 |
| Module 2 | 1 | 0 |
| Module 7 | 1 | 0 |
| Module 8 | 1 | 0 |
| TC respected | 8388 <= 14,400 | 0 <= 14,400 |

**Table 20.** Simulation for belt conveyor station #2 with a sales percentage below 100%.

|  | Assembly Station 1 | Assembly Station 2 |
|---|---|---|
| Cost | 1 | 10 |
| Module 3 | 1 | 0 |
| Module 4 | 1 | 0 |
| Module 5 | 1 | 0 |
| Module 6 | 1 | 0 |
| Module 9 | 1 | 0 |
| TC respected | 6984 <= 14,400 | 0 <= 14,400 |

It is therefore possible to produce all the modules while respecting the cycle time by having only one station per assembly station (station #1 and station #2), requiring the presence of one employee at each station.

By simulating the order of each module at the same time and taking into account that two stations are available according to the module sale percentage, we obtain the results in Tables 21–23 for the feeder conveyors.

**Table 21.** Simulation for feeder conveyor position #1 with a 100% sales percentage.

|  | Assembly Station 1 | Assembly Station 2 |
|---|---|---|
| Cost | 1 | 10 |
| Module 1 | 1 | 0 |
| Module 3 | 1 | 0 |
| TC respected | 9000 <= 14,400 | 0 <= 14,400 |

**Table 22.** Simulation for feeder conveyor station #2 with a sales percentage below 100%.

|  | Assembly Station 1 | Assembly Station 2 |
|---|---|---|
| Cost | 1 | 10 |
| Module 2 | 1 | 0 |
| Module 4 | 1 | 0 |
| TC respected | 4500 <= 14,400 | 0 <= 14,400 |

**Table 23.** Simulation for feeder conveyor station #2 with a sales percentage below 100%.

|  | **Assembly Station 1** | **Assembly Station 2** |
|---|---|---|
| Cost | 1 | 10 |
| Module 1 | 1 | 0 |
| Module 2 | 1 | 0 |
| Module 3 | 1 | 0 |
| Module 4 | 1 | 0 |
| TC respected | 13,500 <= 14,400 | 0 <= 14,400 |

It is therefore possible to produce all the modules, respecting the cycle time, with only one station, requiring the presence of only one employee.

### 4.3. Adapt the Business Model to 4.0

To adapt a business model to 4.0, we must attempt to increase the agility and interconnectivity of the sales and after-sales service, as illustrated in Figure 6. Online sales through e-commerce and predictive maintenance through sensors added on the modules make it possible to significantly reduce reaction time both at the production level and at the follow-up level after installation at the customer's site.

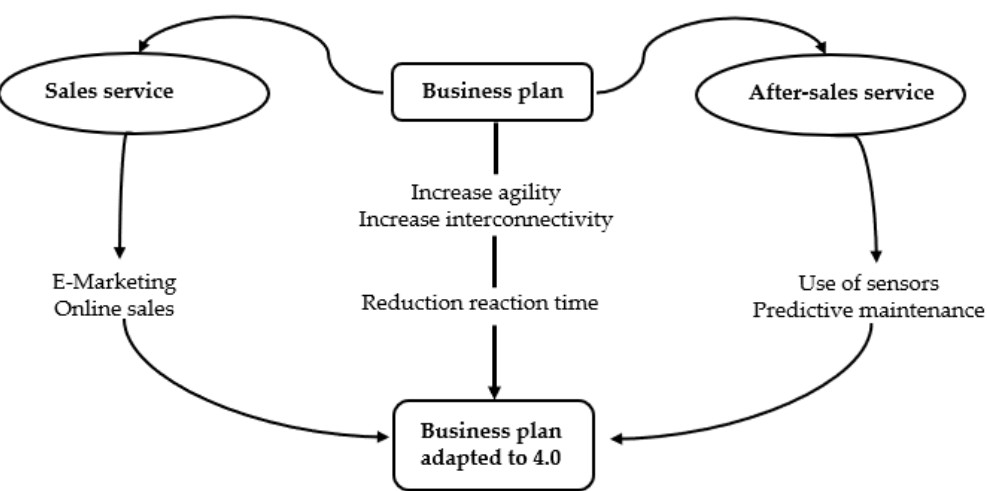

**Figure 6.** Approach for adopting the 4.0 business model.

### 4.3.1. Adapt Marketing Model

The initial purchase order had to be adapted to the modules. This work made it possible to significantly simplify purchase orders. Then, with the implementation of an adapted ERP software, it was possible to move towards an online order form. E-commerce reduces reaction time by allowing us to react in real time. When customers place an order, it is immediately forwarded to the logistics office to be scheduled and placed in production. Moreover, by having standard modules and access to a three-dimensional visual of the components, customers can visualize their order and create it directly online.

With e-commerce, it is also possible to make updated versions of products available in real time. A single version is therefore available to both customers and dealers. Furthermore, online sales make it possible to know when a potential customer is shopping on the site. Adding the real-time chat option online increases proximity between the buyer and the seller and allows for better customer service. It also makes it possible to anticipate business opportunities by following potential customers in real time.

### 4.3.2. Predictive Maintenance

Based on the work of Cimini et al. [27], it can be observed that the target SME is a firm with an "innovative" servitization strategy that seeks to increase the capacity of its products and jointly offer the related service. The company is trying to implement a "solutionist" strategy by optimizing product capacity and selling a complete solution to a need [21].

In order to move towards a more data-driven predictive maintenance strategy, one must be able to target the critical parameters to be studied. The Failure Modes, Effects, and Criticality Analysis (FMECA) method is used to analyze possible failures, their causes, and their effects on the mechanism. The FMECA makes it possible to operationalize the predictive maintenance after-sales service offered with the product. As shown in Tables 24 and 25, knowing the ways in which conveyor functions can be altered makes it easier to know which parameters to monitor. In addition, it is easier to address the product's prospects by having a clearly defined plan for what and how often maintenance should be done.

**Table 24.** FMECA for belt conveyors.

| Module | Equipment | Function | Failure Mode | Consequence | Detection Mode | Criticality Calculation | | | | Satisfied or Not | Recommendation |
|---|---|---|---|---|---|---|---|---|---|---|---|
| | | | | | | G | F | D | C | | |
| 1 | Roller | Transmit motion to the belt. | Degraded operation. Failure in operation. | Belt stops. | Wear and tear | 2 | 1 | 3 | 6 | Undesirable | Addition of sensors to check critical parameters (number of revolutions, temperature). |
| 7 | Roller | Transmit motion to the belt. | Degraded operation. Failure in operation. | Belt stops. | Wear and tear | 2 | 1 | 3 | 6 | Undesirable | Addition of sensors to check critical parameters (number of revolutions, temperature). |
| 8 | Gearbox | Reduce motor speed. Increase torque. | Degraded operation. Failure in operation. | Machine stops running. | Leakage | 2 | 2 | 3 | 12 | Unacceptable | Addition of sensors to check critical parameters (vibration, temperature, oil level). |
| 8 | Electric motor | Transform electrical energy into mechanical energy. Generate rotation of the roller. | Degraded operation. Failure in operation. | Machine stops running. | Leakage | 2 | 2 | 3 | 12 | Unacceptable | Addition of sensors to check critical parameters (vibration, temperature, oil level). |

**Table 25.** FMECA for feeder conveyors.

| Module | Equipment | Function | Failure Mode | Consequence | Detection Mode | Criticality Calculation | | | | Satisfied or Not | Recommendation |
|---|---|---|---|---|---|---|---|---|---|---|---|
| | | | | | | G | F | D | C | | |
| 3 | Gearbox | Reduce motor speed. Increase torque. | Degraded operation. Failure in operation. | Machine stops running. | Leakage | 2 | 2 | 3 | 12 | Unacceptable | Addition of sensors to check critical parameters (vibration, temperature, oil level). |
| 3 | Motor | Transform electrical energy into mechanical energy. Generate rotation of the roller. | Degraded operation. Failure in operation. | Machine stops running. | Leakage | 2 | 2 | 3 | 12 | Unacceptable | Addition of sensors to check critical parameters (vibration, temperature, oil level). |

Integrating sensors will make it possible to read data such as vibration, temperature, oil level, and the number of revolutions of the various pieces of equipment. By targeting the critical values of the components subject to maintenance, it is possible to dispatch a technician on site to perform equipment maintenance even before a breakdown or a failure occurs, while avoiding unnecessary preventive maintenance. A single team can perform both installation and maintenance. These sensors not only allow real-time data acquisition to considerably reduce reaction time in the event of a breakdown or failure but also to increase equipment performance by being aware of operating parameters in the field. By deploying a maintenance team in advance of potential breakdowns, it is possible

to guarantee quality after-sales service and to better predict the distribution of technicians around the world. Moreover, this service can be included at the time of sale.

## 5. Results and Discussion

The global implementation strategy proposed in this study to increase the agility of a Quebec manufacturing SME in the context of the fourth industrial revolution involves three interrelated steps: standardizing the product into modular structures, promoting the linear physical flow and the information flow, and adapting the business model. By increasing the agility of an SME, it is possible to be able to better respond to a customized mass demand.

One of the objectives of this paper was to study the effect of modular product design grouped on platforms on the success of implementing agility and I4.0. To achieve this, historical sales data and product characteristics were analyzed. New modular product designs were created. In this case study, the standardization and modularization of products allowed us to reduce by 70% the components related to belt conveyors and to reduce by 63% the components related to feeder conveyors. Not only is the number of components reduced but also the inventory level, the risk of non-compliance, and the module assembly time, all the while increasing the ease of assembly. Finally, there is currently a crucial period between the customer's order and the factory shipment. By having standard modules ready or assembled in parallel, reaction time can be reduced. It also eliminates the need to start design from zero with every demand, drastically reducing reaction and delivery times. Standard modules increase business agility, making Industry 4.0 implementation easier.

Another objective of this paper was to make the process 4.0 by promoting linear physical flow and information flow. In order to achieve this, a coding scheme adapted to the modular design had to be implemented. Following this, it was possible to revise the inventory management method used and to set up dynamic work cells. In parallel to these steps, an ERP system was implemented. In this case, the work of codifying standard modules facilitated module traceability and tracking. A problem had to be solved here. ERP did not support module management. We had to adapt the system. Implementing an ERP system allowed task planning to be performed in finite capacity and implementing level 3 Kanbans made it possible to significantly reduce throughput time. Then, implementing dynamic cells made it possible to perform assembly tasks on a single workstation. Improving the physical linear flow as well as the information flow at the production level increases the company's agility, making it more prepared for the reality of Industry 4.0.

Finally, the last objective of this paper was to adapt the business model to 4.0 by improving sales and after-sales service. This is accomplished by developing an e-commerce model and prioritizing predictive maintenance. To adapt the business model to 4.0, the customer service process had to be revised both at the sales level and at the post-sales follow-up level. In this case, transitioning purchase orders from an Excel file to a direct online order made it possible to considerably reduce reaction time and to see business opportunities coming. Integrating sensors to the modules also reduces reaction time in the event of a breakdown and increases the service offered. Improving sales and after-sales service reduced reaction time and increased the company's agility, making it more competitive in a I4.0 environment.

At the end of this study, several elements stood out as success factors. Figures 7 and 8 bridge the gap between the obstacles encountered during this case study and the human and technical success factors that had to be put in place to overcome the obstacles.

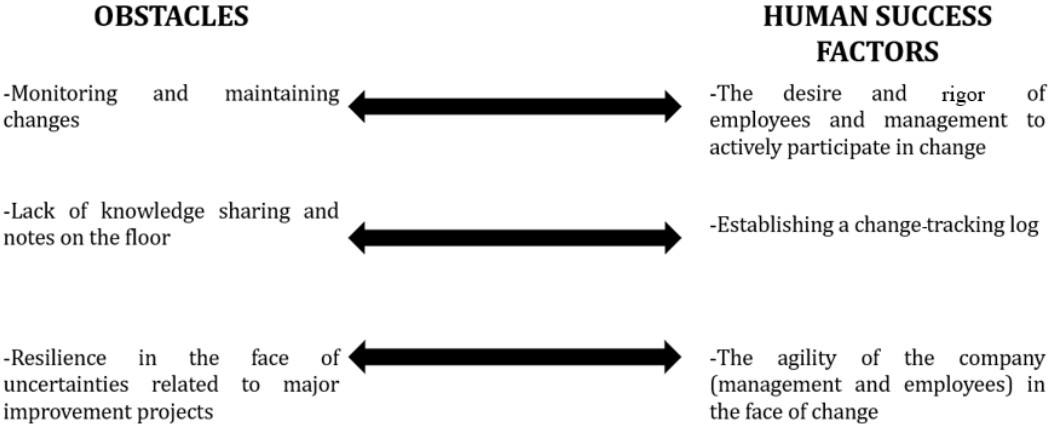

**Figure 7.** Human success factors.

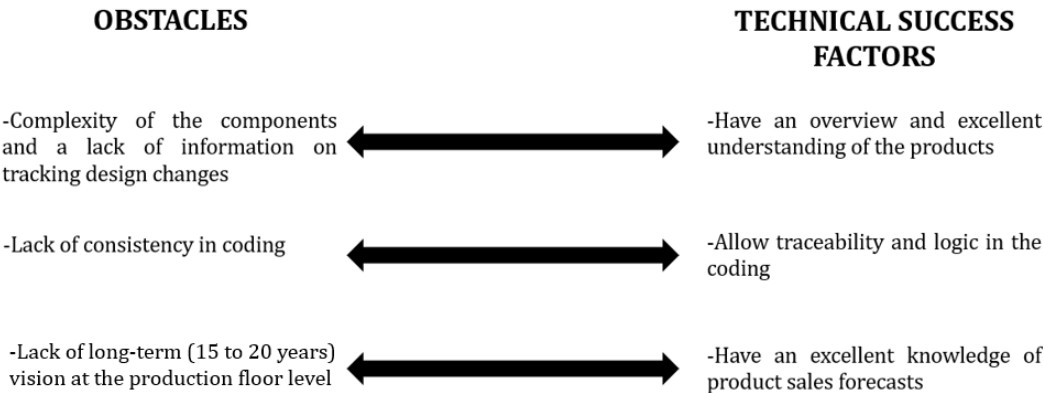

**Figure 8.** Technical success factors.

As Sony and Naik [32] highlighted through their research, the importance to align Industry 4.0 initiatives with organizational strategy, to have the senior management fully supporting the project, to have the participation of employees, and to make a smart product and a smart service was found as an important element of this research. To successfully implemented agility and Industry 4.0, it is essential to put in place these critical factors.

Among the various success factors put forward, the following were essential to the successful implementation of a 4.0 strategy in manufacturing SME:

- Employee and management willingness to actively participate in change: It was surprising to see the willingness of employees to have someone take the time to design the conveyors. It is often reported that one of the biggest barriers to change is employee reluctance. Here, the employees wanted to be involved in change.
- Company rigor and agility in the face of change: It is essential to maintain the changes; continue to improve the product, process, and business model; and continually adapt to the competition in order to be competitive in a I4.0 environment.
- An excellent knowledge of products, history, and sales forecasts: Design review is much easier and more efficient with a good knowledge of the product. It is also easier to anticipate problems and fix them before they happen and to plan the strategy to implement.

The research leads to the conclusion that the available ERP software are not suitable for product modularity. To overcome this issue, the product configurator offers a solution that needs to be further explored. The addition of a configurator allows for better support of product modularity and, by the same token, allows for the operationalization of standardization in a modular structure.

In the same vein as Abdulnour [9], this research shows the importance of increasing and developing I4.0 implementation strategies better adapted to manufacturing SME in a context of personalized mass production.

## 6. Conclusions

This research allowed us to propose a strategy for implementing I4.0 in Quebec manufacturing SME by adapting the product, the process, and the business model. This strategy made it possible to significantly reduce reaction time and to increase agility, connectivity, and ultimately the company's performance to make it competitive in the context of the fourth industrial revolution. Furthermore, this research has led to the conclusion that it is possible to work in parallel on the product, the process, and the business model in order to quickly reap more benefits.

The work performed as part of this study made it possible to standardize easily reconfigurable modules for a specific type of product, namely conveyors. However, we should not limit ourselves to a single product. Every product within companies should be standardized into modules in order to be brought into the I4.0 revolution. This work offers a strategy for implementing I4.0 in Quebec's small- and medium-sized manufacturing companies in three main steps: making the product, the process, and the business model 4.0. This strategy also helps to develop the agility of SME. It would also be interesting to study the possibility of using robots for the welding and painting stages to increase performance and profitability in a manufacturing company. Future research can be done in other sectors of the economy. We are working now to adapt the strategy to a distributive manufacturing system, also known as a network system. Two networks are under study.

Mass customization in the SME context is an issue of growing concern and needs to be further explored, as there is currently no established methodology to achieve this. It would therefore be interesting as future research to develop and validate a mass customization implementation strategy.

**Author Contributions:** S.B.: Conceptualization, methodology, software, validation, formal analysis, investigation, data curation, original draft preparation, visualization; G.A.: Conceptualization, methodology, validation, review and editing, supervision, project administration; S.G.: Conceptualization, methodology, validation, review and editing, supervision. All authors have read and agreed to the published version of the manuscript.

**Funding:** This research received no external funding.

**Institutional Review Board Statement:** Not applicable.

**Informed Consent Statement:** Not applicable.

**Data Availability Statement:** Not applicable.

**Conflicts of Interest:** The authors declare no conflict of interest.

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
