# Peer review of "Agility and Industry 4.0 Implementation Strategy in a Quebec Manufacturing SME"

_sustainability, doi:10.3390/su14137884_

Round 1

Reviewer 1 Report

the paper does not have the characteristics of a research article at all, it is a description of a recorded solution implemented in one enterprise. The lack of a research gap means that no analysis of the literature on this subject is carried out, as if only after key words. It is hard to figure out what this analysis is for the authors. They pass smoothly without binding to methods, but so to invoke any scientific method. Description of the selection of the sample or the reasons for choosing this particular enterprise. The results are a description of the solutions used, but it is difficult to interpret whether these are future plans or already existing solutions. There are also no evaluations of these solutions, perhaps by employees or perhaps by the company's clients? I also don't see the possibility of future research.

Author Response

We want to thank you and the reviewers for your valuable comments. We answered all the reviewers’ comments, one by one, and we added the missing references. Please see our comments in red. The reviewers’ comments made the paper even better as well as more complete and more precise.

This study is part of a larger research program. Four case studies have been or are being conducted. These case studies will complement the results of this research shortly. This case study conducted by Bouchard in a manufacturing SME demonstrated that pairing Lean with 4.0 reduced the number of parts in inventory by 70% and cut makespan by almost  70%. Gamache et al.  presented a case where modular product structure, Lean and 4.0 were implemented and the company’s revenues rose from $2.5 million to $7.5 million in three years with the same resources. Today the company has revenues of $20 million with only three times the previous number of employees. Abdul-Nour, S. et al. demonstrated that the same strategy in a third SME led to a 20% rise in productivity when applying Lean principles and another 20% by implementing some 4.0 tools. Implementing Lean and agility leads to successful I4.0 implementation.

The other case studies were conducted in this sector and a fourth is ongoing. These additional studies will allow us to refine the I4.0 implementation strategy in SME. However, our results already show to SMEs that appreciable gains are possible by implementing Lean 4.0, using the proposed strategy.

Other sectors have been targeted for our future studies. Thus, our results can be generalized to other sectors of activity. Following the encouraging results concerning Lean 4.0 in this study, other case studies will be conducted.

Furthermore, a research project is ongoing to apply these same principles to distributed manufacturing systems, known as network companies, specialized in personalized mass production. Two networks are currently under study.

Reviewer 2 Report

The paper has a very practical topic of creating a methodology for implementing I4.0 into praxis. However, I have a couple of comments and issues with the manuscript:

  • I do not like using just "4.0" in the title of the paper as it should give a clear idea about the topic of the paper ... and "4.0" has many to no meaning.
  • The authors highlight the necessity of agility, flexibility and performance for I4.0, which I do agree with. However, there are also different parameters such as safety and (cyber-)security [A, B], which markedly impact the implementation process itself.
  • The literature review seems selective and excludes the available scientific review papers focusing on the implementation of the I4.0 such as [C,D,E,F], which add as well for example socio-technological layer to the issue of implementing I4.0, key factors of implementing I4.0, company patterns and others.
  • Fig. 6 could use some numbers, i.e., what is "long-term vision" ? 1 3 5 7 10 15 20 years? Moreover, I am missing some comparisons with state-of-the-art. I expect in the literature some systematic review as well on the topic of the paper, which will give a clear picture of the novelty of the paper. Also, highlighting the state of the art and generalization of the finding by the authors might in the end give a clear picture of where is the added value more than just showing an example of implementation, which is not bad, but I still would expect generalization over the current state of the art.

[A] FUJDIAK, R.; et al. Seeking the Relation between Performance and Security in Modern Systems: Metrics and Measures. In 41st International Conference on Telecommunications and Signal Processing (TSP). International Conference on Telecommunications and Signal Processing (TSP). 2018. s. 288-293. ISBN: 978-1-5386-4695-3. ISSN: 1805-5435.

[B] FUJDIAK, R.; et al. Security and Performance Trade-offs for Data Distribution Service in Flying Ad-Hoc Networks. In 2019 11th International Congress on Ultra Modern Telecommunications and Control Systems and Workshops (ICUMT). Ireland, Dublin: 2019. s. 1-5. ISBN: 978-1-7281-5763-4.

[C] DA SILVA, Vander Luiz, et al. Implementation of Industry 4.0 concept in companies: Empirical evidences. International Journal of Computer Integrated Manufacturing, 2020, 33.4: 325-342.

[D] HOYER, Christian; GUNAWAN, Indra; REAICHE, Carmen Haule. The implementation of industry 4.0–a systematic literature review of the key factors. Systems Research and Behavioral Science, 2020, 37.4: 557-578.

[E] DAVIES, Robert; COOLE, Tim; SMITH, Alistair. Review of socio-technical considerations to ensure successful implementation of Industry 4.0. Procedia Manufacturing, 2017, 11: 1288-1295.

[F] SONY, Michael; NAIK, Subhash. Critical factors for the successful implementation of Industry 4.0: a review and future research direction. Production Planning & Control, 2020, 31.10: 799-815.

Author Response

(The authors gave the same response as above.)

Reviewer 3 Report

The article proposes an implementation strategy for agility and Industry 4.0 in manufacturing SME to meet dynamic demand and customized mass production. A single case study is used as the methodology to validate the implementation strategy.

The article starts with a short introduction followed by a literature review section. The structure of the paper should be described at the end of the introduction.

Section 2 Literature review does not include a description of the literature review methodology. How was the literature review carried out? What methodology was used to search and select relevant literature? 

Section 3. Methodology needs to be improved. The authors state that  Based on the literature review, an implementation strategy based on adapting the product, the production process and the business model was developed. The development of this implementation strategy has not been described. How did the authors decide on the three steps of the methodology described in Table 1? It is not clear how these steps result from the literature review.

The authors state the following (lines 238-239) A case study is the methodology used in this research to develop and validate the implementation strategy. The case study design has not been described in the methodology section. How was the single case study company selected? How does it represent Quebec manufacturing SMEs?

Section 5. Discussion does not reflect how the results compare to previous research in the field. This needs to be improved.

Section 6. Conclusions includes figures 5 and 6. These seem to present results of the research rather than conclusions. How were the human and technical success factors studied?

The article title and several sentences in the text use 4.0 probably to instead of Industry 4.0. Why did the authors decide to use this type of abbreviation?

Author Response

(The authors gave the same response as above.)

Reviewer 4 Report

This is an interesting paper on IR 4.0 implementation which is timely and relevant in the time of pandemic and beyond.

  1. Title should not start with a number, maybe reword the title.
  2. 2. The abstract is not informative need to expand and also include some findings and the implications.
  3. The introduction needs to build up a case why this research is warranted currently in the broader context of SMEs.
  4. The Industry 4.0 discussion should cover a broader context of SMEs so that what has already been done can be learnt before deciding on the factors and changes needed.
  5. Currently the focus is too narrow as such has missed out many newer literature on the same issue.
  6. The methodology section needs more detail in terms of selection and description and why certain changes were made etc.
  7. Discussion needs to be more detailed as currently very superficial.
  8. Implications to the firm and beyond is needed as to what can be learnt after the study.
  9. The references need to be checked for format and completeness.

Author Response

(The authors gave the same response as above.)

Round 2

Reviewer 1 Report

What is described in the first review has still not been taken into account. As a reviewer, I am not interested in the history of this study or the project it comes from. I am interested in the research contribution to the manuscript, which still takes away from the standards of this journal.

Author Response

Thank you for your comments.

Please loock at the yellow sections ti fond the answer to your comments.

We Appreciate.

best regards.

Reviewer 3 Report

Literature review methodology description is still missing. How was the literature search carried out, which keywords were used, which databases were utilized?

The case study design description is still missing. How was the single case study company selected? How does it represent Quebec manufacturing SMEs? According to authors' response lines 157-190 in the revised version address this, but this addition concerns literature, not case study design.

Discussion still does not reflect how the results compare to previous research in the field. No references to previous research are mentioned.

Author Response

thank you for your comments. we appreciate.

Please loock to our answer to your comments in yellow.

Best Regards.

Round 3

Reviewer 1 Report

article corrected in line with the comments

Reviewer 3 Report

Earlier review comments have been taken into account.